

# Greenland Ice Sheet discharge from 2000 to 2018

Kenneth D. Mankoff[1], Willam Colgan[1], Anne Solgaard[1], Nanna B. Karlsson[1], Andreas P. Ahlstrøm[1], Dirk van As[1], Jason E. Box[1], Shfaqat Abbas Khan[2], Kristian K. Kjeldsen[1], Jeremie Mouginot[3], and Robert S. Fausto[1]

[1]Department of Glaciology and Climate, Geological Survey of Denmark and Greenland (GEUS), Copenhagen, Denmark
[2]DTU Space, National Space Institute, Department of Geodesy, Technical University of Denmark, Kgs. Lyngby, Denmark
[3]Department of Earth System Science, University of California, Irvine, CA, USA

**Correspondence:** Ken Mankoff (kdm@geus.dk)

**Abstract.** We present a new 18-year (2000 to 2018) estimate of Greenland Ice Sheet ice discharge. Our data include all ice that flows faster than 100 m yr⁻¹ and are generated through an automatic and adaptable method, as opposed to conventional hand-picked gates. We position gates near the present-year termini and estimate problematic bed topography (ice thickness) values where necessary. In addition to using annual time-varying ice thickness, our time series uses velocity maps that begin
with monthly estimates and sparse spatial coverage and ends with ~6-day estimates and near-complete spatial coverage. The recent average (2010 to 2017) ice discharge through the flux-gates is ~500 ± 50 Gt yr⁻¹. The 10 % uncertainty stems from uncertain ice bed location (ice thickness). We attribute the ~50 Gt yr⁻¹ differences among our results and previous studies to our use of updated bed topography from BedMachine v3. Discharge increases from 2000 to 2005, then appears approximately steady. However, regional variability is more pronounced, with decreases at all major discharging glaciers and in all but one
sector offset by increases in the NW sector. As part of the journal's living archive option, all input data, code, and results from this study will be updated when new input data are accessible and made freely available at https://doi.org/doi:10.22008/promice/data/ice_discharge.

## 1 Introduction

The mass of the Greenland ice sheet is decreasing (e.g. Fettweis et al. (2017); van den Broeke et al. (2017); Wiese et al. (2016);
Khan et al. (2016)). Most ice sheet mass loss – as iceberg discharge, submarine melting, meltwater runoff, and basal ablation – enters the fjords and coastal seas, and therefore ice sheet mass loss directly contributes to sea-level rise (WCRP Global Sea Level Budget Group, 2018; Moon et al., 2018; Nerem et al., 2018; Chen et al., 2017). Greenland's total ice loss can be estimated through a variety of independent methods, for example 'direct' mass change estimates from GRACE (Wiese et al., 2016) or by using satellite altimetry to estimate surface elevation change, which is then converted into mass change (using a
firn model, e.g. Khan et al. (2016)). However, attributing the mass loss into iceberg discharge (D) and surface mass balance (SMB) remains challenging (c.f. Rignot et al. (2008) and Enderlin et al. (2014)). Correctly assessing mass loss, as well as the attribution of this loss (SMB or D) is critical to understanding the process-level response of the Greenland ice sheet to climate change, and thus improving models of future ice-sheet changes and associated sea-level rise (Moon et al., 2018).





The total mass of an ice-sheet, or a drainage basin, changes if the mass gain (SMB inputs, primarily snowfall) is not balanced by the mass loss (D and SMB outputs, generally meltwater runoff). This change is typically termed ice-sheet mass balance (MB) and the formal expression for this rate of change in mass is (e.g. Cuffey and Paterson (2010)),

$$\frac{\mathrm{d}M}{\mathrm{d}t} = \rho \int_A \dot{b}\,\mathrm{d}A - \int_g Q\,\mathrm{d}g,$$ (1)

where $\rho$ is the density of ice, $b$ is the surface mass balance, and $Q$ is the volumetric flow rate. The left hand side of the equation is the rate of change of mass, the first part on the right hand side is the area integrated surface mass balance (SMB), and the second term is the volumetric flow rate that drains through gate $g$, or discharge (D). Equation 1 is often simplified to

$$MB = SMB - D$$ (2)

and referred to as the "input-output" method (e.g. Khan et al. (2015)). Virtually all studies agree on the trend of Greenland

mass balance, but large discrepancies persist in both the magnitude and attribution. Magnitude discrepancies include, for example, Kjeldsen et al. (2015) reporting a mass imbalance of -250 $\pm$ 21 Gt yr[-1] during 2003 to 2010, Ewert et al. (2012) reporting -181 $\pm$ 28 Gt yr[-1] during 2003 to 2008, and Rignot et al. (2008) reporting a mass imbalance of -265 $\pm$ 19 Gt yr[-1] during 2004 to 2008. Some of this difference may be due to different ice sheet area masks used in the study. Attribution discrepancies include, for example, Enderlin et al. (2014) attributing the majority (64 %) of mass loss to changes in SMB

during the 2005 to 2009 period but Rignot et al. (2008) attributing the majority (85 %) of mass loss to changes in D during the 2004 to 2008 period.

Discharge may be calculated through several methods, including volumetric flow rate through gates (e.g. Enderlin et al. (2014); King et al. (2018)), or solving as a residual from independent mass balance terms (e.g. Kjær et al. (2012); Kjeldsen et al. (2015)). The gate method that we use in this study incorporates ice thickness and an estimated vertical velocity profile

from the observed surface velocity to calculate the discharge. The term discharge refers to the ice volumetric flow rate at or close to the grounding line. A typical formulation of discharge across a gate $D_g$ is,

$$D_g = \rho V H w,$$ (3)

where $\rho$ is depth-averaged density, $V$ is depth-averaged gate-perpendicular velocity, $H$ is the ice thickness, and $w$ is the width. Uncertainties in $V$ and $H$ naturally influences the estimated discharge. At fast-flowing outlet glaciers, $V$ is typically

assumed to be equal at all ice depths, and observed surface velocities can be directly translated into depth-averaged velocities (as in Enderlin et al. (2014); King et al. (2018)). To avoid added uncertainty from SMB or basal process corrections downstream of a flux gate, the gate should be at the grounding line of the outlet glacier. Unfortunately, uncertainty in radar-derived bed elevation (translating to ice thickness uncertainty) increases toward the grounding line.





Conventional methods of gate selection involve hand-picking gate locations, generally as linear features (e.g. Enderlin et al. (2014)) or visually approximating ice-orthogonal gates at one point in time (e.g. King et al. (2018)). Manual gate definition is suboptimal. For example, the largest discharging glaciers draw from an upstream radially-diffusing region that may not easily be represented by a single linear gate. Approximately flow-orthogonal curved gates may not be flow-orthogonal on the multi-decade time scale due to changing flow directions. Manual gate selection makes it difficult to update gate locations, corresponding with glacier termini retreat or advance, in a systematic and reproducible fashion. We therefore adopt an algorithmic approach to generate gates based on a range of criteria.

Here, we present a time-evolving discharge dataset based on gates selected in a reproducible fashion by a new algorithm. Relative to previous studies, we employ ice velocity observation with higher temporal frequency and denser spatial coverage. Our expanded data set permits estimates of discharge at high frequency from all Greenland ice sheet outlet glaciers. We use ice velocity from 2000 to 2018 including six-day velocities for the last ~500 days of the time series, and discharge at 200 m per pixel resolution capturing all ice flowing faster than 100 m yr$^{-1}$ that crosses glacier termini into fjords.

## 2 Input data

Historically, discharge gates were selected along well-constrained flight-lines of airborne radar data (Enderlin et al., 2014). Recent advances in ice thickness estimates through NASA Operation IceBridge (Millan et al., 2018), NASA Oceans Melting Greenland (OMG; Fenty et al. (2016)), fjord bathymetry (Tinto et al., 2015), and methods to estimate thickness from surface properties (e.g. McNabb et al. (2012); James and Carrick (2016)) have been combined into digital bed elevation models such as BedMachine v3 (Morlighem et al., 2017b, a) or released as independent datasets (Millan et al., 2018). From these advances, digital bed elevation models have become more robust at tidewater glacier termini and grounding lines. The incorporation of flight-line ice thickness data into higher-level products that include additional methods and data means gates are no longer limited to flight-lines (e.g. King et al. (2018)).

Ice velocity data are available with increasing spatial and temporal resolution (e.g. King et al. (2018); Vijay et al. (2019)). Until recently, ice velocity mosaics were limited to once per year during winter (Joughin et al., 2010), and they are still temporally limited to annual resolution prior to 2000 (e.g. Mouginot et al. (2018b)). Focusing on recent times, ice-sheet wide velocity mosaics from the Sentinel 1A & 1B are now available every six days (http://PROMICE.org). The increased availability of satellite data has improved ice velocity maps both spatially and temporally thereby decreasing the need to rely on spatial and temporal interpolation of velocities from annual/winter mosaics (Andersen et al., 2015; King et al., 2018).

The discharge gates in this study are generated using only surface speed and an ice mask. We use the MEaSUREs Greenland Ice Sheet Velocity Map from InSAR Data, Version 2 (Joughin et al., 2010, 2015, updated 2018), hereafter termed "MEaSUREs 0478" due to the National Snow and Ice Data Center (NSIDC) date set ID number. We use the BedMachine v3 (Morlighem et al., 2017b, a) ice mask.

For ice thickness estimates, we use surface elevation from GIMP (Howat et al. (2014, 2017); NSIDC data set ID 0715), adjusted through time with surface elevation change from Khan et al. (2016). We use bed elevations from BedMachine v3





replaced by Millan et al. (2018) where available. Ice catchment delineation is from Mouginot et al. (2017) and Rignot and Mouginot (2012) (see Supplemental Material). Ice velocity data are obtained from a variety of products including Sentinel 1A & 1B derived by PROMICE (see Supplemental Material for Sentinel velocity information), MEaSUREs 0478, and MEaSUREs 0646 (Howat, 2017). Official glacier names come from Bjørk et al. (2015). Other glacier names come from Rignot et al. (2008). See Table 1 for an overview of data sets used in this work.

Our compilation includes 235 different velocity maps, biased toward the last 500 days of the time series when six-day ice velocities become available from the Sentinel-1 satellites. The temporal distribution (apparent from the plots) is 9 to 13 velocity maps per year from 2000 through 2015, 24 in 2016, and 55 in 2017.

**Table 1.** Summary of data sources used in this work. First column is the physical property covered by the data. Second column is the informal name used in this work to reference this data source. Third column is(are) the reference(s).

| Property | Name used in this paper | Reference |
|---|---|---|
| Basal Topography | BedMachine | Morlighem et al. (2017b, a) |
| Basal Topography for Southeast | | Millan et al. (2018) |
| Surface Elevation | GIMP 0715 | Howat et al. (2014, 2017) |
| Surface Elevation Change | Surface Elevation Change | Khan et al. (2016) |
| Velocity | Sentinel | Supplemental Material |
| Velocity | MEaSUREs 0478 | Joughin et al. (2015, updated 2018) |
| Velocity | MEaSUREs 0646 | Howat (2017) |
| Catchments & Sectors | Catchments & Sectors | Supplemental Material |
| Names | | Bjørk et al. (2015); Rignot et al. (2008) |

## 3 Methods

### 3.1 Terminology

We use the following terminology, most displayed in Fig. 1:

– "Pixels" are individual 200 m x 200 m raster discharge grid cells.

– "Gates" are contiguous (including diagonal) clusters of pixels.

– "Catchments" are spatial areas that have 0, 1, or > 1 gate(s) plus any upstream source of ice that flows through the gate(s), and come from Mouginot et al. (2017) and Rignot and Mouginot (2012) (See Supplemental Material).

– "Sectors" are groups of catchments, also from Mouginot et al. (2017), and labeled by approximate geographic region.

– The "baseline" period is the average 2015, 2016, and 2017 winter velocity from MEaSUREs 0478.



- "Coverage" is the percentage of total, sector, catchment, or gate discharge observed at any given time. By definition coverage is 100 % during the baseline period. From the baseline data, the contribute to total discharge of each pixel is calculated, and coverage is reported for all other maps that have missing observations (Fig. A2). Total estimated discharge is always reported because missing pixels are gap-filled (see "Missing and invalid data" section below).

– "Fast-flowing ice" is defined as ice that flows more than 100 m yr$^{-1}$.

## 3.2  Gate location

Gates are algorithmically generated for fast-flowing ice (greater than 100 m yr$^{-1}$) close to the ice sheet terminus determined by the baseline-period data. We define the termini using the BedMachine ice mask. We buffer termini 5000 m in all directions and once again only select fast-flowing ice pixels. Our procedure results in gates 5000 m upstream from the baseline terminus that
bisect the baseline fast-flowing ice. We manually mask some land- or lake-terminating glaciers which are initially selected by the algorithm due to fast flow and mask issues.

We select a 100 m yr$^{-1}$ speed cutoff because slower ice, taking longer to reach the terminus, is more influenced by SMB. For the influence of this threshold on our results see the Discussion section below and Fig. 2.

We select gates at 5000 m upstream from the baseline termini, which means that gates are likely > 5000 m from the termini
further back in the historical record (Murray et al., 2015; Wood et al., 2018). The choice of a 5000 m buffer follows from the fact that a low-as-possible distance-to-terminus is desirable to avoid the need for (minor) SMB corrections downstream, yet is not too close to the terminus where discharge results are sensitive to the choice of distance-to-terminus value (Fig. 2), which may be indicative of bed (ice thickness) errors.

## 3.3  Discharge

We calculate discharge per pixel using density (917 kg m$^{-3}$), ice speed from satellite imagery, pixel width, and ice thickness derived from time-varying surface elevation and a fixed bed elevation (Eq. 3). We assume that any change in surface elevation corresponds to a change in ice thickness and thereby neglect basal uplift, erosion, and melt, which combined are orders of magnitude less than surface melting (e.g. Cowton et al. (2012); Khan et al. (2007)). We also assume depth-averaged ice velocity is equal to the surface velocity.

## 3.4  Missing and invalid data

The baseline data provides velocity at all gate locations by definition (Fig. 3), but individual non-baseline velocity maps often have missing data. Also, thickness provided by BedMachine is clearly incorrect in some places (e.g. fast-flowing ice that is 10 m thick, Fig. 4). We define invalid data and fill in missing data as described below.





### 3.4.1 Missing velocity

We generate an ice speed time series by assigning the PROMICE, MEaSUREs 0478, and MEaSUREs 0646 products to their respective reported time stamps (even though these are time-span products). We ignore that any individual velocity data set (map) and even point (pixel) has a time span, not a time stamp. Velocities are sampled only where there are gate pixels. Missing

pixel velocities are linearly interpolated in time, except for missing data at the beginning or end of the time series which are backward- and forward- filled (respectively) with the temporally-nearest value for that pixel (Fig. A2). We do not spatially interpolate missing velocities because the spatial changes around a missing data point are most likely larger than the temporal changes. We visually represent the discharge contribution of directly observed pixels, termed coverage (Fig. A2) as time series graphs and opacity of dots and error bars in the figures. Therefore, the gap-filled discharge contribution at any given time is

equal to 100 minus the coverage. Discharge is always reported as estimated total discharge even when coverage is less than 100 %.

### 3.4.2 Invalid thickness

We derive thickness from surface and bed elevation. We use GIMP 0715 surface elevations in all locations, and the BedMachine bed elevations in most locations, except southeast Greenland where we use the Millan et al. (2018) bed. The GIMP 0715 surface

elevations are all time-stamped per pixel. We adjust the surface through time by linearly interpolating elevation changes from Khan et al. (2016), which covers the period from 2000 to 2016. We linearly interpolate the average of the last three years to later times. Finally, from the fixed bed and temporally varying surface, we calculate the time-dependent ice thickness at each gate pixel.

The thickness data generated as described above (Fig. 5) are unlikely to be valid at all locations. For example, many locations

have fast-flowing ice (Fig. 3), but report ice thickness as 10 or less m (Fig. 4, left panel). We accept all ice thickness greater than 20 m and construct from this a thickness versus $\log_{10}$ speed relationship. For all ice thickness less than or equal to 20 m thick (at each pixel location) we adjust thickness based this relationship (Figs. 5 and 4, right panel). We selected the 20 m thickness cutoff after visually inspecting the velocity distribution (Fig. 4, right panel). This thickness adjustment adds 21 Gt to our baseline-period discharge estimate with no adjustment. In the Supplemental Material and Table A2 we discuss the

discharge contribution of these adjusted pixels, and a comparison among this and other thickness adjustments.

### 3.4.3 Ice Discharge Uncertainty

We estimate the uncertainty related to the ice discharge following a simplistic approach. This yields an uncertainty of the total ice discharge of approximately 10 % throughout the time series.

At each pixel we estimate the maximum discharge, $D_{\max}$, from

$$D_{\max} = \rho\left(V + \sigma_V\right)\left(H + \sigma_H\right)W, \tag{4}$$



and minimum discharge, $D_{\min}$, from

$$D_{\min} = \rho\,(V - \sigma_V)\,(H - \sigma_H)\,W, \tag{5}$$

where $\rho$ is ice density, $V$ is velocity, $\sigma_V$ is the velocity error, $H$ is ice thickness, $\sigma_H$ is the ice thickness error, and $W$ is the width at each pixel. Included in the thickness term is surface elevation change through time ($\mathrm{d}H/\mathrm{d}t$) and its uncertainty

($\sigma_{\mathrm{d}H/\mathrm{d}t}$). However, because $\sigma_H \gg \sigma_V$ and $\sigma_H \gg \sigma_{\mathrm{d}H/\mathrm{d}t}$, both $\sigma_V$ and $\sigma_{\mathrm{d}H/\mathrm{d}t}$ terms are ignored. When data sets do not come with error estimates we treat the error as 0.

On a pixel by pixel basis we used the provided thickness uncertainty for each dataset. Where we modified the thickness (H < 20 m), we prescribe an uncertainty of 0.5 times the adjusted thickness. Subsequently, the uncertainty on individual glacier-, catchment-, sector-, or ice sheet scale is obtained by summarizing, but not reducing by the square of the sums, the uncertainty

related to each pixel. An in-depth discussion related to treatment of errors and uncertainty is in the Supplementary Information.

## 4 Results

### 4.1 Gates

Our discharge gate algorithm generates 5981 pixels making up 264 gates, assigned to 172 ice-sheet sectors following Mouginot et al. (2017). Previous similar studies have used 230 gates (King et al., 2018) and 178 gates (Enderlin et al., 2014).

The widest gate (~47 km) is Sermersuaq (Humboldt Gletsjer), the 2nd widest (~34 km) is Sermeq Kujalleq (Jakobshavn Isbræ). 23 additional glaciers have gate lengths longer than 10 km. The minimum gate width is 3 pixels (600 m) by definition in the algorithm.

The average thickness at unadjusted gates is 407 m with a standard deviation of 259. The average thickness at gates with erroneous values adjusted is 440 m with a standard deviation of 226. A histogram of unadjusted and adjusted thickness at all

gate locations is shown in Fig. 5, speed at all gate locations is shown in Fig. 3, and a combined 2D histogram in Fig. 4.

### 4.2 Ice discharge (volumetric flow rate)

Our ice discharge dataset (Fig. 6) reports a total discharge of $468 \pm 47$ in 2000 and $515 \pm 50$ Gt/yr in 2005, after which annual discharge remains approximately steady at 495 to $520 \pm {\sim}50$ Gt/yr during the 2005 to 2017 period. Annual maxima in ice discharged occurred in 2005 ($515 \pm 50$ Gt/yr), 2011 ($521 \pm 53$ Gt/yr), and 2013 ($519 \pm 53$ Gt/yr). Discharge in both 2016 and

2017 was less than 500 Gt each year.

At the sector scale, the SE glaciers (see Fig. 1 for sectors) are responsible for 148 to 169 ($\pm$ 12 %) Gt yr[-1] of discharge (42 to 54 % of ice-sheet wide discharge) over the 2007 to 2017 period. By comparison, the predominantly land-terminating NO, NE and SW together were responsible for only $\sim$70 Gt yr[-1] of discharge (32 % of ice-sheet wide discharge) during this time (Fig. 7). The discharge from most sectors has been approximately steady or declining for the past decade. The NW is the only

sector exhibiting a persistent increase in discharge; From ~90 to 115 Gt yr[-1] (22 % increase) over the 2000 to 2018 period (+





Gt yr$^{-1}$ or + 0.9 % yr$^{-1}$). This persistent increase in NW discharge is offsetting declining discharge from other regions. The largest contributing sector, SE, contributed a high of 175 ± 20 Gt in 2003, but dropped to 148 (154) ± 18 Gt in 2016 (2017). The last time the SE sector was persistently below 150 Gt yr$^{-1}$ was in the early 2000s.

In the NO, NE and SW sectors, which contribute a minority of ice-sheet discharge, low coverage (large data gaps) is evident in the coverage chart (Fig. 7, only NO of these three sectors is shown for clarity), and as linear trends with data point centers and error bars transparent. These indicators of gap-filling are also evident in the NW sector but only in 2014 & 2015. They are more clearly evident in the NE, NO, and SW with adjusted scaling (Fig. C1 in Supplemental Material is the same as Fig. 7 but with a logarithmic y-axis).

Focusing on the top seven contributors at the individual glacier scale (Fig. 8), Sermeq Kujalleq (Jakobshavn Isbræ) has slowed down from an annual average high of ~55 Gt yr$^{-1}$ in 2012 to ~45 Gt yr$^{-1}$ in 2016 and ~38 Gt yr$^{-1}$ in 2017. The 2012 to 2016 slowdown of Sermeq Kujalleq (Fig. 8) is compensated by the many glaciers that make up the NW sector (Fig. 7). The large 2017 reduction in discharge at Sermeq Kujalleq is partially offset by a large increase in the 2nd largest contributor, Helheim Gletsjer (Fig. 8).

### 4.3 Volumetric flux

Thinning and accelerating ice may balance each other to maintain a steady volume flow rate, but the same thinning and accelerating ice would increase flux due to the decreased cross-sectional flux area. Since the flux is proportional to gate size (height) and ice velocity, it is highly dependent on gate location. Volume flow rate is roughly steady since 2005, and increased ~12 % between the 2001 minimum (461 Gt yr$^{-1}$) and 2013 maximum (521 Gt$^{-1}$). During this same period the flux increased 16 % from a 2001 minimum (0.80 Gt yr$^{-1}$ km$^{-2}$) to a 2011 maximum (0.95 Gt yr$^{-1}$ km$^{-2}$). Flux exhibits a similar year-to-year variability as discharge (volume flow rate) but the flux signal is overlaid on a continuously accelerating trend line, equal to the inverse of the thinning rate.

### 5 Discussion

Different ice discharge estimates among studies likely stem from three categories: 1) changes in true discharge, 2) different input data (ice thickness and velocity), and 3) different assumptions and methods used to analyze data. Improved estimates of true discharge is the goal of this and many other studies, but changes in true discharge (category 1) can happen only when a work extends a time series into the future because historical discharge is fixed. Thus, any inter-study discrepancies in historical discharge must be due to category 2 (different data) or category 3 (different methods). Most studies use both updated data and new or different methods, but do not always provide sufficient information to disentangle the two. This is inefficient. To more quantitatively discuss inter-study discrepancies, it is imperative to explicitly consider all three potential causes of discrepancy. Only when results are fully reproducible – meaning all necessary data and code are available (c.f. Rezvanbehbahani et al. (2017)) – can new works confidently attribute discrepancies relative to old works. Therefore, in addition to providing new




discharge estimates, we attempt to examine discrepancies among our estimates and other recent estimates. Without access to code and data from previous studies, it is challenging to take this examination beyond a qualitative discussion.

The algorithm-generated gates we present offer some advantages over traditional hand-picked gates. Our gates are shared publicly, are generated by a code that can be audited by others, and are easily adjustable within the algorithmic parameter space. This adjustability allows both sensitivity testing of gate location (Fig. 2) and allows gate positions to systematically evolve with glacier termini (not done here, because we also report flux which is sensitive to gate location, in addition to flow rate). The total ice discharge we estimate is ~10 % less than the total discharge of two previous estimates (Enderlin et al., 2014; Rignot et al., 2008), and similar to that of King et al. (2018), who attributes their discrepancy with Enderlin et al. (2014) to the latter using only summer velocities, which have higher annual average values than seasonally-comprehensive velocity products. The gate locations also differ among studies, and glaciers with baseline velocity less than 100 m yr$^{-1}$ are not included in our study due to our velocity cutoff threshold, but this should not lead to substantially different discharge estimates (Fig. 2).

Our gate selection algorithm also does not place gates in northeast Greenland at Storstrømmen, Bredebræ, or their confluence, because during the baseline period that surge glacier was in a slow phase. We do not manually add gates at these glaciers. The last surge ended in 1984 (Reeh et al., 1994; Mouginot et al., 2018a), prior to the beginning of our time series, and these glaciers are therefore not likely to contribute substantial discharge even in the early period of discharge estimates.

We instead attribute the majority of our discrepancy with Enderlin et al. (2014) to the use of differing bed topography in southeast Greenland. When we compare our top ten highest discharging glaciers in 2000 with those reported by Enderlin et al. (2014), we find that the Køge Bugt discharge reported by Enderlin et al. (2014) is ~31 Gt, but our estimate is only ~16 Gt (and ~17 Gt in King et al. (2018)). The Bamber et al. (2013) bed elevation dataset most likely employed by Enderlin et al. (2014) has a major bed depression in central Køge Bugt bed. This region of enhanced ice thicknesses is not present in the BedMachine dataset that we and King et al. (2018) employ (Fig. B1). If the Køge Bugt gates of Enderlin et al. (2014) are in this location, then those gates overlie Bamber et al. (2013) ice thicknesses that are about twice those reported in BedMachine v3. With all other values held constant, this results in roughly twice the discharge. Although we do not know whether BedMachine or Bamber et al. (2013) is more correct, conservation of mass suggests that a substantial subglacial depression should be evident as either depressed surface elevation or velocity (Morlighem et al., 2016).

We are unable to attribute the remaining discrepancy between our discharge estimates and those by Enderlin et al. (2014), but agree with King et al. (2018) that differing seasonal velocity sampling may be the cause.

Our ice discharge estimates agree well with the most recently published discharge data (King et al., 2018), with one notable difference. The King et al. (2018) 2005 peak discharge is 524 ± 9 Gt dropping to 461 ± 9 Gt in 2008 – a decrease of ~63 Gt. In our work, the 2005 peak is 515 ±50 Gt, dropping to 495 ± 50 Gt in 2008 – a decrease of only ~20 Gt. We note that our uncertainty estimates include the King et al. (2018) estimates, but the opposite does not appear be true. We suggest the discrepancy in 2005 to 2008 discharge decrease results from differing approaches to temporal interpolation during a period of highly transient discharge. King et al. (2018) use seasonally varying ice thicknesses, derived from seasonally varying surface elevations, and a Monte Carlo method to temporally interpolate missing velocity data to produce discharge estimates.



In comparison, we use linear interpolation of both yearly surface elevation estimates and temporal data gaps. It is not clear whether linear or higher-order statistical approaches are best-suited for interpolation during the non-linear discharge changes seen between 2005 and 2008. Our use of unmodified velocity products (except for linear gap-filling) also highlights some of the imperfections in those products. It is not visible in the total discharge graph, but when viewing individual glaciers (Fig. 8) the signal to noise ratio decreases, and any individual outlying data point should be treated with caution.

We calculate the gate-orthogonal velocity at each pixel and at each timestamp, meaning all velocity estimates are gate-orthogonal at all times, regardless of gate position, orientation, or changing glacier velocity direction over time. It is unlikely that discharge estimates using gates that are only approximately flow-orthogonal and time-invariant (King et al., 2018) have large errors due to this, because it is unlikely that glacier flow direction changes significantly, but this treatment may be the cause of some differences among our approach and other works. Discharge calculated using non-orthogonal methodology would overestimate true discharge.

## 6  Data Availability

This work in its entirety is available at doi:10.22008/promice/data/ice_discharge (Mankoff, 2019a). The glacier-scale, catchment, region, and Greenland summed ice sheet discharge dataset is available at doi:10.22008/promice/data/ice_discharge/d/v0. 0.1 (Mankoff, 2019c), where it will be updated as more velocity data become available. The gates can be found at doi:10.22008/ promice/data/ice_discharge/gates/v0.0.1 (Mankoff, 2019d), the code at doi:10.22008/promice/data/ice_discharge/code/v0.0. 1 (Mankoff, 2019b), and the surface elevation change at doi:10.22008/promice/data/DTU/surface_elevation_change/v1.0.0 (Khan, 2017).

## 7  Conclusions

We have presented a novel dataset of flux gates and 2000 to 2018 glacier-scale ice discharge estimate for the Greenland ice sheet. These data are underpinned by an algorithm that both selects gates for ice flux and then computes ice discharges.

From our discharge estimate we show that over the past ~20 years, ice sheet discharge rose to just over 500 Gt yr$^{-1}$ from 2000 to 2005, and has held roughly steady since 2005 at near 500 Gt yr$^{-1}$. However, when viewed at a sector or glacier scale, the system appears more dynamic with spatial and temporal increases and decreases canceling each other out to produce the more stable ice sheet discharge. We note that there does not appear to be any dynamic connection among the sectors, and any increase in one sector that was offset by a decrease in another has likely been due to chance. If in coming years changes occur and the signals happen to have matching signs, then ice sheet discharge will decrease or increase, rather than remain fairly steady.

The application of our flux-gate algorithm shows that ice-sheet wide iceberg discharge varies by ~50 Gt yr$^{-1}$ between the minimum and maximum of the upstream buffer distance (i.e. distance between the flux gates and the glacier termini) and the lateral velocity cut-off of flux gates (Fig. 2). This variance – due only to gate position and shape – is approximately equal to the



uncertainty associated with ice-sheet wide discharge estimates reported in many studies (e.g. Rignot et al. (2008); Andersen et al. (2015); Kjeldsen et al. (2015)). The ice discharge we present here is similar to recent estimates by King et al. (2018) where our time series overlap. We highlight a major discrepancy with the ice discharge data of Enderlin et al. (2014) and we suspect this discharge discrepancy – most pronounced in southeast Greenland – is associated with the choice of digital bed

elevation model, specifically a deep hole in the Bamber et al. (2013) bed at Køge Bugt.

The flux gates, discharge data, and the algorithm used to generate the gates, discharge, and all figures, are freely available. This publication aims to take advantage of this ESSD journal "living data" process to maintain an evolving data set. Transparency in data and methodology are critical to move beyond a focus of estimating discharge quantities, towards more operational mass loss products with realistic errors and uncertainty estimates. The convention of devoting a critical paragraph,

or even page, to methods now appears to be insufficient given the complexity and pace of Greenland ice sheet research. We hope that the flux gates, data, and code we provide here is a step toward helping others both improve their work and discover the errors in ours.

*Acknowledgements.* Author Contribution: KDM conceived of the algorithm approach, and wrote the code. KDM , WIC, and RSF iterated over the algorithm results and methods. ASO provided the velocity data. SAK supplied the surface elevation change data. JM provided the

catchments. All authors contributed to the scientific discussion, writing, and editing of the manuscript. The authors declare that they have no conflict of interest. Data from the Programme for Monitoring of the Greenland Ice Sheet (PROMICE) were provided by the Geological Survey of Denmark and Greenland (GEUS) at http://www.promice.dk. Parts of this work were funded by the INTAROS project under the European Union's Horizon 2020 research and innovation program under grant agreement No. 727890.



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



**Figures**

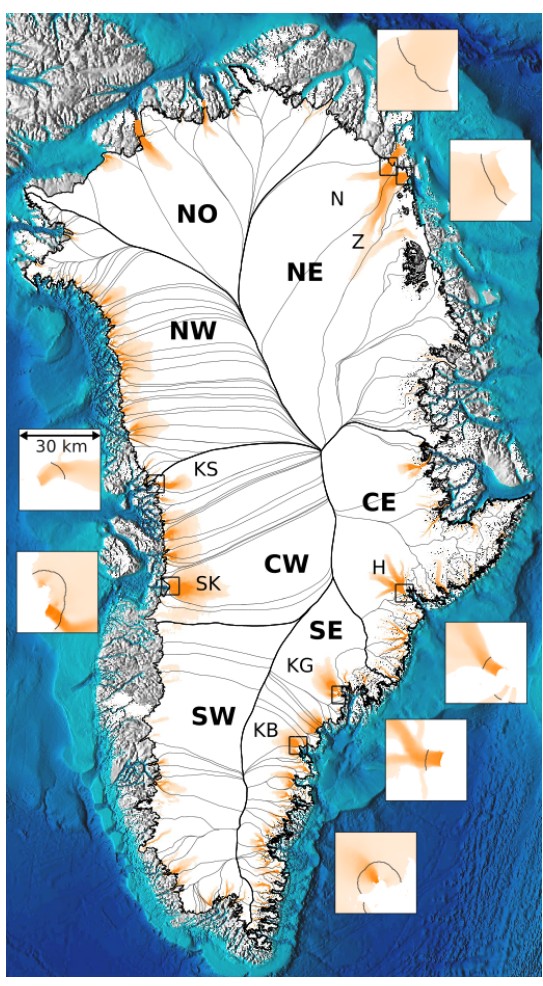

**Figure 1.** Overview showing fast-flowing ice (orange, greater than 100 m yr$^{-1}$) and the gates for the seven top discharging glaciers (Fig. 8). Gates are shown as black lines in inset images. Each inset is 30 x 30 km and all have the same color scaling, but different than the main map. Insets pair with nearest label and box. On the main map, sectors from Mouginot et al. (2017) are designated by thicker black lines and large bold labels. Catchments (same source) are delineated with thinner gray lines, and the top discharging catchments are labeled with smaller font. H = Helheim Gletsjer, KB = Køge Bugt, KG = Kangerlussuaq Gletsjer, KS = Kangilliup Sermia (Rink Isbræ), N = Nioghalvfjerdsbræ, SK = Sermeq Kujalleq (Jakobshavn Isbræ), and Z = Zachariæ Isstrøm. Basemap terrain (gray), ocean bathymetry (blues), and ice mask (white) come from BedMachine.



**Figure 2.** Heatmap and table showing ice sheet discharge as a function of gate buffer distance and ice speed cutoff. The colors of the numbers change for readability.



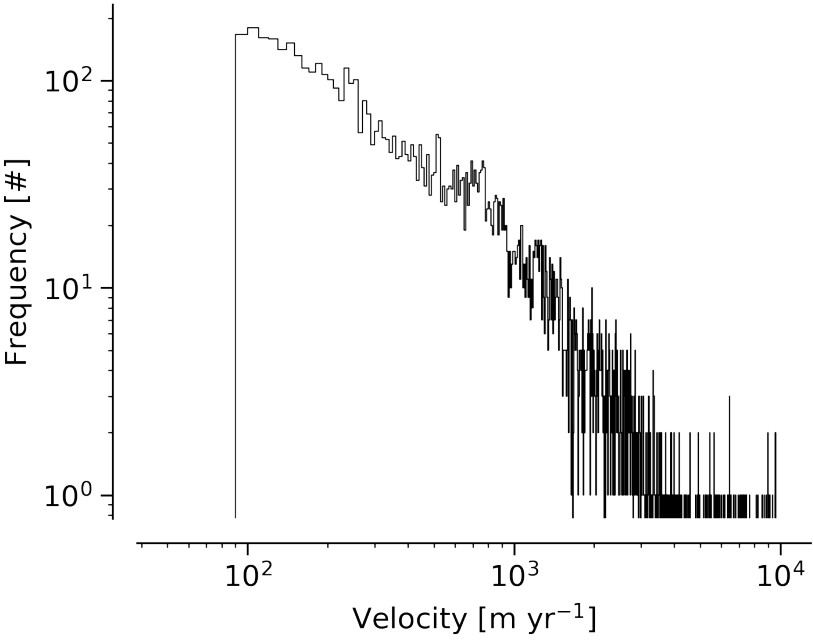

**Figure 3.** Histogram of velocity at gate pixels.

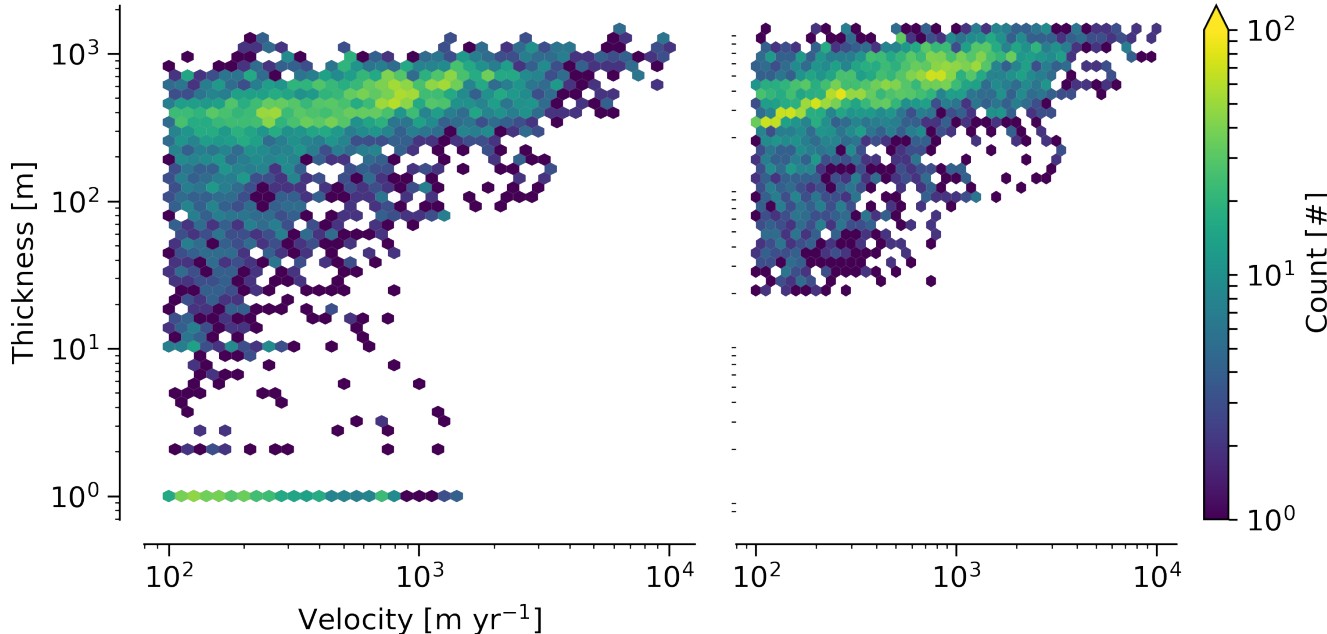

**Figure 4.** 2D histogram of velocity and thickness at all gate pixels. Left panel: Unadjusted (BedMachine & Millan et al. (2018)) thickness. Right panel: Adjusted (as described in the text) thickness.



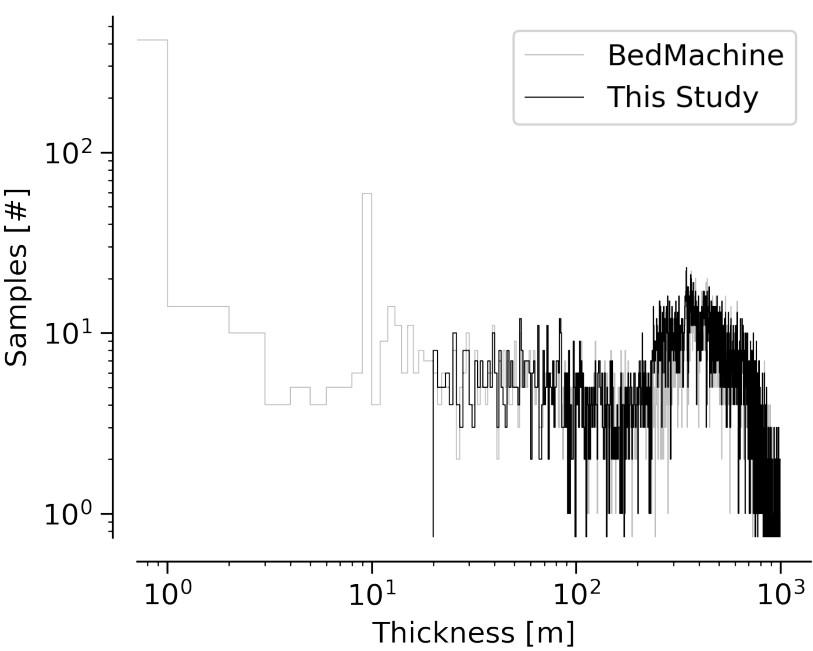

**Figure 5.** Histogram of thickness at gate pixels showing both unadjusted (BedMachine & Millan et al. (2018)) and adjusted (this study) thickness as described in the text.





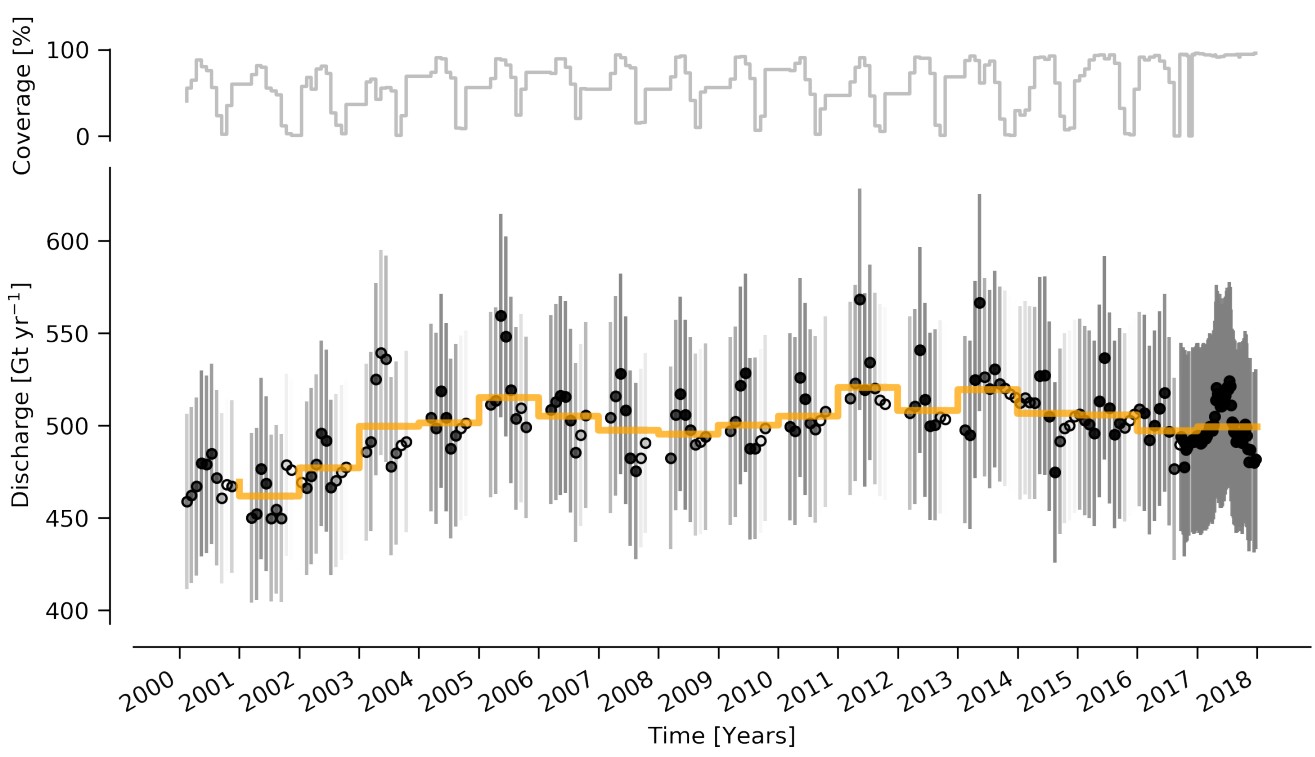

**Figure 6.** Bottom panel: Time series of ice discharge from the Greenland ice sheet. Dots represent when observations occurred. Orange stepped line is annual average. Coverage (percentage of total discharge observed at any given time) is shown in top panel, and also by opacity of dot interior and error bars on lower panel. When coverage is < 100 %, total discharge is estimated and shown by linearly interpolating missing coverage.



**Figure 7.** Bottom panel: Time series of ice discharge by sector. Same graphical properties as Fig. 6. Top panel: The sector with highest coverage (CE), lowest coverage (NE), and coverage for sector with highest discharge (SE) are shown. Coverage for other sectors not shown to reduce clutter.





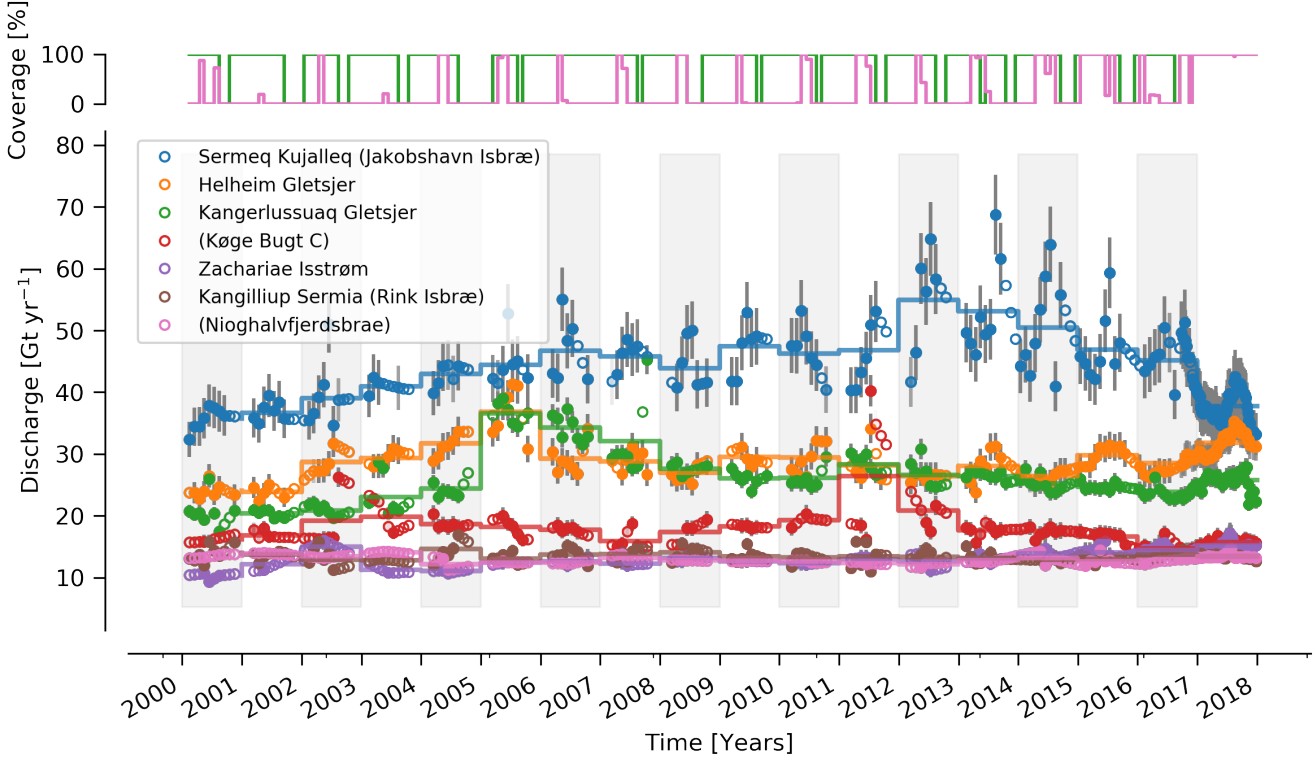

**Figure 8.** Bottom panel: Time series of ice discharge showing top seven discharging glaciers. Same graphical properties as Fig. 6. Only highest (Kangerlussuaq Gletsjer) and lowest (Nioghalvfjerdsbræ) coverage shown to reduce clutter.



## Appendix A: Errors and Uncertainties

Here we describe our error and uncertainty treatments. We begin with a brief philosophical discussion of common uncertainty treatments, our general approach, and then the influence of various decisions made throughout our analysis, such as gate location and treatments of unknown thicknesses.

Traditional and mathematically valid uncertainty treatments divide errors into two classes: systematic (bias) and random. The primary distinction is that systematic errors do not decrease with more samples, and random errors decrease as the number of samples or measurements increases. The question is then which errors are systematic and which are random. A common treatment is to decide that errors within a sector are systematic, and among sectors are random. This approach has no physical basis - two glaciers a few 100 m apart but in different sectors are assumed to have random errors, but two glaciers 1000s of km

apart but within the same sector are assumed to have systematic errors. In reality, it may be the case that all glaciers $<W$ km wide, or $>D$ m deep, have systematic errors even if some are on opposite sides of the ice sheet, at least when the ice thickness is estimated with the same method (i.e. the systematic error is likely caused by the sensor and airplane, not the location of the glacier).

The decision to have $R$ random samples (where $R$ is the number of sectors, usually ~18 based on the Zwally sectors (Zwally

et al., 2012)) is also arbitrary. Mathematical treatment of random errors means that even if the error is 50 %, 18 measurements reduces it to only 11.79 %.

This effect is unlikely to be physically meaningful. Our 171 sectors, 263 gates and 5980 pixels means that even if errors were 100 % for each, we could reduce it to 7.6, 6.2, or 1.3 % respectively. We note that the area error introduced by the common EPSG:3413 map projection is +8 % in the north and -6 % in the south, and while it may be considered in other works, it is not

explicitly mentioned.

We do not have a solution for the issues brought up here, except to discuss them explicitly and openly so that those, and our own, error treatments are clearly presented and understood to likely contain errors themselves.

## A1   Invalid Thickness

We assume ice velocities are correct and ice thicknesses < 20 m are incorrect where ice speed is > 100 m yr$^{-1}$. Of 5980 pixels,

5380 have valid thickness, and 600 (10 %) have invalid thickness. However, the speed at the locations of the invalid thicknesses is generally much less (and therefore the assumed thickness is less), and the influence on discharge is less than an average pixel with valid thickness (Table A1).

When aggregating by gate, there are 263 gates. Of these, 186 (70 %) have no bad pixels and 77 (30 %) have some bad pixels, 53 have > 50 % bad pixels, and 49 (19 %) are all bad pixels.

We adjust these thickness using a poor fit (correlation coefficient: 0.3) of the $\log_{10}$ of the ice speed to thickness where the relationship is known (thickness > 20 m). We set errors equal to one half the thickness (i.e. $\sigma_H = \pm 0.5\,H$). We also test the sensitivity of this treatment to simpler treatments, and have the following four categories:

**NoAdj**  No adjustments made. Assume BedMachine thickness are all correct.



**Table A1.** Statistics of pixels with and without valid thickness. Numbers represent speed [m yr$^{-1}$] except for the "count" row.

|        | Good Pixels | Bad Pixels |
|--------|-------------|------------|
| count  | 5380        | 600        |
| mean   | 820         | 269        |
| std    | 1039        | 230        |
| min    | 100         | 101        |
| 25 %   | 230         | 130        |
| 50 %   | 487         | 178        |
| 75 %   | 970         | 286        |
| max    | 10044       | 1423       |

**NoAdj+Millan**  Same as NoAdj, but using Millan et al. (2018) thickness where available.

**300**  If a gate has some valid pixel thicknesses, set the invalid thicknesses to the minimum of the valid thicknesses. If a gate has no valid thickness, set the thickness to 300 m.

**400**  Set all thickness < 50 m to 400 m

5 **Fit**  Use the thickness v. speed relationship described above.

Table A2 shows the estimated baseline discharge to these four treatments:

**Table A2.** Effect of different thickness adjustments on discharge

| Treatment    | Discharge [Gt] |
|--------------|----------------|
| NoAdj        | $472 \pm 49$   |
| NoAdj+Millan | $480 \pm 49$   |
| 300          | $487 \pm 49$   |
| 400          | $493 \pm 51$   |
| Fit          | $491 \pm 50$   |

Finally, Figure A1 shows the geospatial locations, concentration, and speed of gates with and without bad pixels.

## A2  Missing Velocity

The velocity products come with their own uncertainty value at each location. Here we clarify our temporal gap-filling of
10 missing velocities.

We estimate discharge at all pixel locations for any time when there exists any velocity product. Not every velocity product provides velocity estimates at all locations, and we fill in where there are gaps by linear interpolating velocity at each pixel in time. We calculate coverage, the discharge-weighted percent of observed velocity at any given time (Figure A2), and display

**Figure A1.** Gate locations and thickness quality. Left: locations of all gates. Black dots represent gates with 100 % valid thickness pixels, blue with partial, and red with none. Top right: Percent of bad pixels in each of the 264 gates, arranged by region. Bottom panel: Average speed of gates. Color same as left panel.

coverage as 1) line plots over some of the time series graphs, 2) opacity of the error bars and infilling of time series dots. Linear interpolation and discharge-weighted coverage is illustrated in Figure A2, where pixel A has a velocity value at all three times, but pixel B has a filled gap at time $t_3$. The concentration of valid pixels is 0.5, but the weighted concentration, or coverage, is 9/11 or ~0.82. When displaying these three discharge values, $t_1$ and $t_4$ would have opacity of 1 (black), and $t_3$ would have
5   opacity of 0.82 (dark gray).

This treatment is applied at the pixel level and then averaged to the gate, catchment, sector, and ice sheet results.



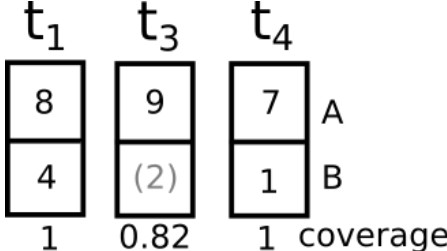

**Figure A2.** Schematic demonstrating coverage. Velocities are filled with linear interpolation in time, and coverage is weighted by discharge. $t$ columns represent the same two gate pixels (A & B) at three time steps, where $t_n$ are linearly spaced, but $t_2$ is not observed anywhere on the ice sheet and therefore not included. Numbers in boxes represents example discharge values. Gray parenthetical number is filled, not sampled, in pixel B at time $t_3$. Weighted filling computes the coverage as $9/11 = 0.\overline{81}$, instead of 0.5 (half of the pixels at time $t_3$ have observations).

## A3   Errors from map projection

Our work takes place in a projected coordinate system (EPSG 3413) and therefore errors are introduced between the "true" earth spheroid (which is itself an approximation) and our projected coordinates system. We address these by calculating the projection error due to EPSG 3413 which is approximately +8 % in Northern Greenland and -6 % in Southern Greenland, and multiplying variables by a scaling factor if the variables do not already take this into account. Velocities are "true velocities" and not scaled, but the nominal 200 m gate width is scaled.





## Appendix B: Køge Bugt Bamber

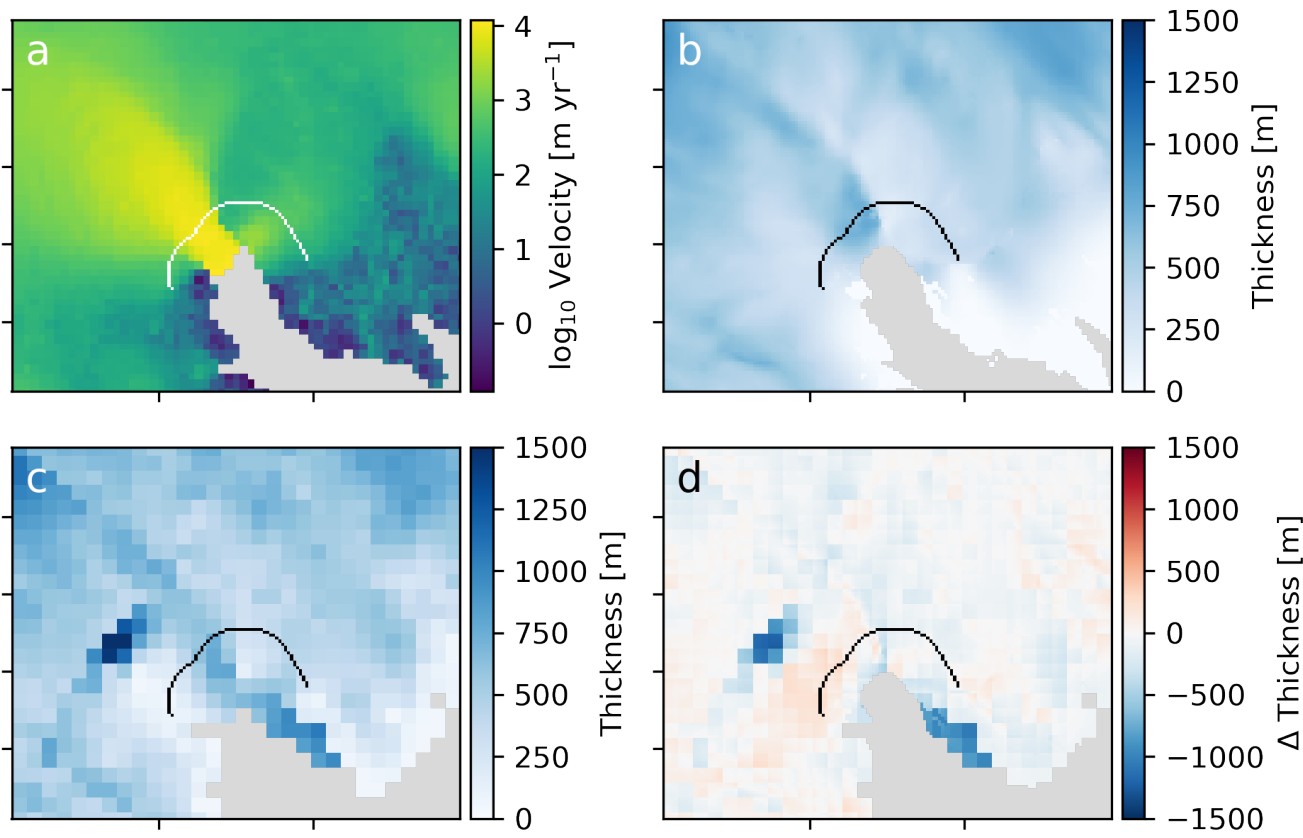

**Figure B1.** Differences between BedMachine and Bamber et al. (2013) near Køge Bugt. Panel (a) is baseline ice speed, (b) BedMachine thickness, (c) Bamber et al. (2013) thickness, and (d) difference computed as BedMachine - Bamber. Curved line is gate used in this work.





## Appendix C: Regional Discharge with logarithmic y-axis

**Figure C1.** Same as Fig. 7 but with logarithmic y-axis so that NE, NO, and SW sectors are more visible





## Appendix D: Sentinel-1 ice velocity maps

We use ESA Sentinel-1 synthetic aperture radar (SAR) data to derive ice velocity maps covering the Greenland Ice Sheet margin using offset tracking (Strozzi et al., 2002) assuming surface parallel flow using the digital elevation model from the Greenland Ice Mapping Project (GIMP DEM, NSIDC 0645) by Howat et al. (2014, 2015). The operational interferometric post processing (IPP) chain (Dall et al., 2015; Kusk et al., 2018), developed at the Technical University of Denmark (DTU) Space and upgraded with offset tracking for ESA's Climate Change Initiative (CCI) Greenland project, was employed to derive the surface movement. The Sentinel-1 satellites have a repeat cycle of 12 days, and due to their constellation, each track has a six-day repeat cycle. We produce a Greenland wide product that spans two repeat cycles of Sentinel-1 A. The product is a mosaic of all the ice velocity maps based on 12 day pairs produced from all the tracks from Sentinel-1 A and B covering Greenland during those two cycles. The product thus has a total time span of 24 days. Six day pairs are also included in each mosaic from track 90, 112 and 142 covering the ice sheet margin in the south as well as other tracks on an irregular basis in order to increase the spatial resolution. (Rathmann et al., 2017) and Vijay et al. (2019) have exploited the high temporal resolution of the product to investigate dynamics of glaciers. The maps are available from 2016-09-13 and onward, are updated regularly, and are freely available from http://promice.dk.

## Appendix E: Catchments and Sectors

Catchment outlines of 260 glaciers are determined using ice flow direction from velocity and from topographic slope. In areas of flow greater than 100 m/yr, we use a composite ice velocity map (Mouginot et al., 2017), and, in slow-moving areas, we use surface slope to get the direction of the flow (GIMP, Howat et al. (2014, 2017)) smoothed over a spatially variable distance of 10 ice-thicknesses to remove short-wavelength undulations of the surface. Glacier catchments are regrouped in 7 large regions, which were defined based on ice flow regime, and the need to divide the ice sheet into areas comparable in size and discharge with relatively homogeneous climate conditions.



**Appendix F: Software**

This work was performed primarily using GRASS GIS (Neteler et al., 2012), Python (Van Rossum and Drake Jr, 1995), and IPython (Pérez and Granger, 2007), in particular the pandas (McKinney, 2010), numpy (Oliphant, 2006), statsmodel (Seabold and Perktold, 2010), x-array (Hoyer and Hamman, 2017), and Matplotlib (Hunter, 2007) libraries. The

5  parallel (Tange, 2011) tool was used to speed up processing. We used proj4 (PROJ contributors, 2018) to compute the errors in the EPSG 3413 projection. All code used in this work is available in the Supplemental Material.