# Peer review of "Greenland Ice Sheet solid ice discharge from 1986 through 2017"

_Earth System Science Data, 2019_

## Referee Comment (RC1) · Ellyn Enderlin (Referee) · 22 Mar 2019

Summary: The authors use a variety of remotely-sensed datasets to develop a new automated method to extract flux gates to map spatio-temporal variations in Greenland glacier discharge, and construct time series of discharge for fast-flowing glaciers draining the ice sheet. They find some discrepancies with previous estimates that are attributed to different datasets and methodologies employed by each study. They also find that the ice sheet discharge has been relatively constant since ∼2005, due to steady or decreasing discharge from most sectors that have been offset by a gradual increase in discharge from the NW portion of the ice sheet. Importantly, the authors have made all data and code available, hopefully leading to easier inter-comparison of future discharge estimates.

[Figure]

The paper is very well written and I really enjoyed some of the clever ways that the data were presented in the figures. Overall, I find that there are only a few minor points that should be addressed in the current version of the manuscript.

Major Comments: 1. Since one of the major arguments the authors make is that the use of flux gates that are picked in an automated way is superior over manually-traced gates, I'd like a bit more thorough description of the method used to pick the flux gates. I follow that you apply a 5000m buffer to all ice inland of the terminus but it is less clear what you mean when you say you "select" fast-flowing ice. Do you essentially place the flux gate at the 100m/yr flow contour? If this is the case, then I imagine that in some regions the flux gate is closer than 5000m from the terminus but the rest of section 3.2 suggests that the gates are a fixed location of 5000m inland of the terminus. The addition of a schematic to illustrate the method would be helpful because the few panels in Figure 1 have arcuate geometries that seem independent of flow speed. 2. For the interpolation/extrapolation of speed and thickness, why did you use linear interpolation techniques? For the speeds, linear interpolation may introduce considerable aliasing effects, particularly if there are large data gaps around times of rapid change (like the peak in speed in the SE in ∼2005). For the thickness data, why did you use the average of the last 3 years with data to estimate thickness for 2017-2018? Was flow relatively steady during this time? Are your results considerably influenced if you would use only the last 1-2 years or expand to include a longer time period? For the speeds you simply use the closest observation at the ends of the time series. Why use a different approach for thickness?

Minor Line-by-Line Comments: p. 3, l. 11: Replace "200 m per pixel" with "200m pixel" p. 6, l. 19-25: This is a clever approach for dealing with the clearly incorrectly thin ice that I have also observed in some locations. p. 7, l. 5: If I understand this correctly, then all of your discharge uncertainty is from thickness uncertainty. Is this correct? p. 7, l. 24: Remove "both" p. 8, l. 3: Is there a particular reason why you use 150Gt/yr as the cutoff here? Is this the estimated balance discharge? p. 8, l. 15-21: The numbers

presented in this section indicate that the flux is the area-normalized volume flow rate. Is this correct? I think that most readers would stumble in this section since it is not apparent from the start that the flux is normalized. I normally think of flux as a volume flow rate and I was perplexed by the seemingly contradictory statements in the first sentence until I looked at the flux units. p. 9, l. 19: The Enderlin et al. (2014) paper used bed elevations from radar picks. Examination of the original interpreted data for Koge Bugt suggest the bed was much deeper than the updated (and BedMachine) dataset. It is likely that the Bamber et al. (2013) bed map used the same radar data as Enderlin et al. (2014). (This comes up again on p. 11, l. 5.) p. 9, l. 26: The use of Khan et al. (2016) surface elevation adjustments may also play a role since the Enderlin et al. (2014) elevations are directly extracted from DEMs and Operation IceBridge lidar timeseries.

---

## Referee Comment (RC2) · Anonymous Referee #2 · 23 Mar 2019

**Summary**

This study calculates Greenland Ice Sheet Discharge for the 20th century by extracting time series for all ice margin pixels that flow at speeds greater than 100 m/yr, comprising 263 discrete flux gates. The methodology is relatively straight forward and the manuscript well-written and easy to follow. The novelty of this work stems from an automated flux gate selection algorithm, which allows the authors to test discharge sensitivity to velocity thresholds and gate-to-terminus distance. The authors have made all data and related code openly-accessible and appear to have the framework in place to continually update the time series through the present. However, I suggest several methodological concerns be addressed prior to publication, primarily regarding the treatment of outliers, temporal averaging, and lack of detail describing the gate selection algorithm itself. Major points are listed first below, followed by minor points and technical comments.

**Major comments:**

Treatment of outliers in velocity data

On Page 10, line 5, the authors state that the signal to noise ratio may be reduced at the individual glacier level but that this noise is not apparent in the total ice sheet discharge time series. I disagree with this assertion and will highlight an example that I think demonstrates the need for a more careful treatment of outliers.

Distinct spikes are apparent in the Greenland total graph (Figure 6) from the ~monthly data in 2011 and 2013. These seem to be primarily the result of similar spikes in the SE sector (Figure 7). In the SE in 2011, there is a rapid fall of ~50 Gt (30%) and a subsequent rise again of ~40 Gt all within a ~3-month period, which is physically improbable. Looking more closely at the top 7 individual glacier plots in Figure 8, we see that the sector spike in 2011 is an artifact of discharge at Køge Bugt Glacier (located in the SE), which increases from a baseline of < 20 Gt/yr to over 40 Gt/yr in 1-2 months. Given the usage of annual surface elevations used in this study, this change must then be due to velocity, and would require an acceleration of over 100% in a very short time period. Below are velocity time series taken from Joughin et al. 2018 (https://doi.org/10.5194/tc-12-2211-2018), which shows no such dramatic acceleration during that time. From my quick glance at the individual glacier discharge data accompanying this manuscript, several other glaciers also occassionally exhibit large, abrupt changes that I suspect are untreated outliers.

[Figure]

Similarly, the next SE sector spike in 2013 is not present in either Køge Bugt or Kangerlussuaq Glacier, which are the largest two glaciers in this sector. This would mean that

the remaining glaciers in the SE sector (which I estimate from the figure to roughly account for an average ~100 Gt/yr) would have to compensate for the ~50 Gt/yr increase. I'm skeptical of this because, to my knowledge, such an abrupt and short-lived acceleration in 2013 has not been previously documented.

These examples show that although signal noise at individual glaciers will typically be mitigated in the total ice sheet time series, large outliers at the more prominent glaciers will propagate to the sector-wide time series and also possibly to the total time series and impact annual averages. The outlier problem is amplified when linear temporal averaging is applied using noisy data points. I think the data quality could be improved by a simple low-pass filter, which would also make the individual glacier time series more robust for those who may use the data for local studies. It may be helpful to reference the velocity maps/mosaics associated with outlying points to assess if the pixels are particular noisy at that time.

Temporal averaging
How are annual averages computed from the nonuniform time series? I suspect that the series are resampled at uniform intervals prior to averaging, but this is not explicitly described in the manuscript. Even if resampled at equal intervals, the use of linear interpolation for missing time periods means that there is an inherent sampling bias that the authors should estimate, though it may be small. This could be done using the reference period with dense temporal coverage.

Lack of text on automated flux gate selection algorithm
I'd like to see more details on the algorithm included in the manuscript, especially since the automated algorithm is the key strength of this work. Some details are commented throughout the code samples, but it requires digging. I suggest a methods section describing the algorithm development that at minimum addresses:
1.) If and how frequent manual adjustments are needed due to continued retreat (terminus retreats behind 5km upstream of GIMP-determined terminus).
2.) The treatment of unconfined, radially-draining catchments – do they require additional corrections as shape and direction of dominant flow change?
3.) Brief description of treatment of floating ice shelves. By termini selection, do you mean grounding ice mask from BedMachine/GIMP or glacier front?
4.) Does the algorithm ever require "unfiltering" originally excluded pixels after gate migration? For example, though most glaciers accelerate toward the terminus, if a pixel 5 km upstream of a slow moving near-terminus pixel exceeds the 100 m/yr threshold, is it still excluded, or retroactively filled along the gate? I suspect this would only happen in some instances at radially-draining glaciers.

**Minor comments**

Title and use of '2000-2018' period.
The figures and the description that the period studied as '18 year' (abstract) indicate that the time series extends through the end of 2017 and excludes data from 2018. While the use of 2000 'to' 2018 could be taken to mean 'up until' 2018, '2000 to 2018' and '20XX to 2017' are interchangeably used throughout the manuscript and it is confusing to the reader which exact period is being referenced. For example, an average discharge from 2010 to 2017 is mentioned on line 6 of the abstract. Based on the usage of the title, is this taken to mean the average was

calculated over the period 2010 through 2016, excluding 2017? Similarly, on page 7, line 27, a sector average is described over the 2007 to 2017 period. If the terminology is consistent and this does indeed refer to 2007 *through* 2016, why is 2017 excluded in these instances? Otherwise, please consider either replacing '2000 to 2018' with '2000 through 2017', or editing the remaining 'to 2017' references for consistency.

Page 2, Equation 1
Perhaps specify that A is area (even though it's intuitive) so that all terms are defined.

Page 2, Line 5
"…and Q is the volumetric flow rate". This makes is sound like Q should be in Equation 1. Are you defining now for later use? If the term is not used again it might be best to omit.

Page 3, Line 2
Contribute should be "contribution"

Page 5, Line 20
The use of 917 kg/m3 density value should be noted again in the discussion when comparing to previous studies as it could be another, albeit small, source of difference.

3.4.1 Missing Velocity
Do the reported stamps refer to the time span midpoint, or the first date of the time span (first image)?

Page 6, Line 23
"This thickness adjustment adds 21 Gt to our baseline-period discharge estimate..."
Should be Gt/yr? Is this adjustment, described in Table A2 and as applied to the final estimates in Figure 6, added as a fixed value to the full time series or does the magnitude of the adjustment vary through time?

3.4.3 Ice Discharge Uncertainty
Is the temporal variability in coverage considered in error estimates? I would expect discharge estimates for a given time with, for example, only ~20% coverage to have a larger uncertainty than a time point with full coverage.

4.2 Ice discharge (volumetric flow rate)
Page 7, line 26
If 169 Gt/yr is 54% of total ice sheet discharge, this yields a total ice sheet discharge of ~313 Gt/yr. On line 28, 70 Gt/yr representing 32% of total ice sheet contribution would indicate that total ice sheet discharge is ~219 Gt/yr. These values are inconsistent with each other and with the preceding paragraph. Can some text be added here to clarify what these percentages represent?

4.3 Volumetric flux
I found this section confusing. By normalizing discharge by cross-sectional area, the authors are effectively describing interannual changes in velocity (since density is constant through time). I

think it would be easier to follow if described in velocity terms, but it may not be necessary to include this section at all.

Page 8, line 18
2013 maximum should be 2011 (as previously stated on page 7, line 24).

Page 9, line 29
"The King et al. (2018) 2005 peak discharge is 524 +/- 9 Gt dropping to 461 +/- 9 Gt in 2008 – a decrease of ~63 Gt. In our work, the 2005 peak is 515 +/- 50 Gt, dropping to 495 +/- 50 Gt in 2008 – a decrease of only ~20 Gt."

This comparison should be altered to either (1) compare annual averages between both studies, or (2) compare absolute max and min value between 2005 and 2008 from the ~monthly estimates (which look to be about 550 - 480=70 Gt from Fig. 6). The annual changes from King et al. 2018, plotted in red in Figure 1 from that paper, show an annual change of ~20 Gt, which is comparable with these results.

Figure 1
This is a very good figure.

Figure 2
The heatmap is an excellent addition to the manuscript and packs a lot of information into a very readable figure. Interesting to see sensitivity to cutoff velocities increases with upstream gate distance.

Figure 8 and discussion of 7 largest discharging glaciers
I accessed the individual glacier discharge data available on the data portal and calculated that 'IKERTIVAQ_M' glacier contributes a period average discharge of ~14 Gt/yr, which is larger than the average contribution from Nioghalvfjerdsbrae. Why was this glacier excluded from the top 7? If instead this is a reference to the top 7 glaciers spread throughout each sector rather than absolute largest 7, then consider adding a word on this for clarification.

Page 24, line 17
Is the gate number 263 or 264? Both values are used throughout the manuscript.

Table A2
Are these values given for the reference period (2015-2017)?

---

## Author Comment (AC1) · 21 Apr 2019

We have replied to all comments from the reviewers. Please see the attached document.

Please also note the supplement to this comment:
https://www.earth-syst-sci-data-discuss.net/essd-2019-29/essd-2019-29-AC1-supplement.pdf

---

## Author Response (AR1)

**Reply to Reviews for "Greenland Ice Sheet discharge from 2000 to 2018"**

**K. D. Mankoff & Co-Authors**

Comments from reviewers are in normal font and differentiated from the replies that use a **bold colored font**. Most changes are tracked in the document showing changes between versions, but latexdiff sometimes fails with reference changes, so many references to the new Mouginot et al. (2019) are not highlighted. In addition, changes to graphics and the bibliography are not highlighted.

This single PDF contains the reply to the reviewers, the revised document highlighting most of the changes, and then the revised document without changes highlighted.

**Contents**

| 1 | Reply to Editor                                                                                                                                     |                       |  |  |  |  |  |  |  |  |  |
|---|-----------------------------------------------------------------------------------------------------------------------------------------------------|-----------------------|--|--|--|--|--|--|--|--|--|
| 2 | 2       Reply to Reviewer #1 Ellyn Enderlin (Referee) doi:10.5194/essd-2019-29-RC         2.1       Major Comments         2.2       Minor Comments |                       |  |  |  |  |  |  |  |  |  |
| 3 | Reply to Reviewer #2 doi:10.5194/essd-2019-29-RC2         3.1       Major Comments                                                                  | 4
4
6
6
7 |  |  |  |  |  |  |  |  |  |
| 4 | 1 References                                                                                                                                        |                       |  |  |  |  |  |  |  |  |  |
| 5 | 5 Revised Document Showing Changes                                                                                                                  |                       |  |  |  |  |  |  |  |  |  |
| 6 | Revised Document                                                                                                                                    |                       |  |  |  |  |  |  |  |  |  |

**1 Reply to Editor**

We are happy to read these reviews due to their general support of our work and constructive suggestions. Our revised text has addressed all of the reviewer comments and we also reply explicitly to each comment below.

We have removed two figures (the velocity and thickness histograms) because the information contained in those is also contained in the 2D velocity v. thickness histogram.

2 Reply to Reviewer #1 Ellyn Enderlin (Referee) doi:10.5194/essd-2019-29-RC1

**2.1 Major Comments**

Since one of the major arguments the authors make is that the use of flux gates that are picked in an automated way is superior over manually-traced gates, I'd like a bit more thorough description of the method used to pick the flux gates. I follow that you apply a 5000m buffer to all ice inland of the terminus but it is less clear what you mean when you say you "select" fast-flowing ice. Do you essentially place the flux gate at the 100m/yr flow contour? If this is the case, then I imagine that in some regions the flux gate is closer than 5000m from the terminus but the rest of section 3.2 suggests that the gates are a fixed location of 5000m inland of the terminus. The addition of a schematic to illustrate the method would be helpful because the few panels in Figure 1 have arcuate geometries that seem independent of flow speed.

We have clarified the text, but to answer the question here, when we "select" fastflowing ice we do not place a gate at a 100 m yr-1 contour - we use a 2D raster mask for all ice with speed > 100 m yr-1. Then, from this subset, we find the ice edge. The ice edge is the contour line only where the fast ice is near the ocean mask, not all segments of the contour where the fast ice is bounded by slow ice, or land. We then buffer the edge 5000 m in all directions (over slow ice, into the fjord or coastal sea, etc.) creating an oval-like shape around each terminus. We then crop this circular-ish feature by the fast ice mask from the previous step. The result is an arc segment from the circular-ish feature that transects only the fast ice. We call this the gate.

For the interpolation/extrapolation of speed and thickness, why did you use linear interpolation techniques? For the speeds, linear interpolation may introduce considerable aliasing effects, particularly if there are large data gaps around times of rapid change (like the peak in speed in the SE in 2005).

We used a linear interpolation to fill in velocity because we opted for simplicity where possible. Aliasing may be introduced, and we now note that in the text. The method used by King et al. (2018) improves on our method in some situations, but we feel it is not appropriate in others – using past-as-predictor in a changing system means the changes may not be treated properly. Specifically, Jakobshavn no longer appears to be following the annual cycle, so a model that is based on statistical monthly behavior may not be appropriate.

For the thickness data, why did you use the average of the last 3 years with data to estimate thickness for 2017-2018? Was flow relatively steady during this time? Are your results considerably influenced if you would use only the last 1-2 years or expand to include a longer time period? For the speeds you simply use the closest observation at the ends of the time series. Why use a different approach for thickness?

We back- and forward-fill speed (extend rather than extrapolate) because speed has a highly variable second derivative so extrapolation will introduce larger errors than extension. That is, extrapolation is sensitive to the last trend, while extension, which effectively assumes steady state, is safer for the points back-filled in the early 1990s.

We used linear for thickness interpolation because that was the best fit among the four relationships (linear &  $log_{10}$  for velocity & speed) we explored. We could get a much better correlation coefficient if we used discharge for the dependent variable as in Enderlin et al. (2014), but we were unable to justify that decision because the relationship is then a form of self-correlation – velocity exists in both terms and they are highly co-dependent, but the correlation coefficient test assumes variables are independent.

We chose to use a 3-year smooth after examining the rate of change of 2-year or 1-year. We chose this as a subjective value. It does not have a large effect on the early part of the time series – rates of change are low in the 90's.

2.2 Minor Comments

p. 3, l. 11: Replace "200 m per pixel" with "200m pixel"

Done.

p. 7, l. 5: If I understand this correctly, then all of your discharge uncertainty is from thickness uncertainty. Is this correct?

Correct. We now explain in the text (and show in the supplemental material) that proportional velocity uncertainty is an order of magnitude less than proportional thickness uncertainty. Adding velocity uncertainty and treating it as random w.r.t. thickness uncertainty would add a small amount to the total uncertainty. Because we are already conservative in our uncertainty estimate, we choose to ignore this small additional uncertainty.

p. 7, l. 24: Remove "both"

Done.

p. 8, l. 3: Is there a particular reason why you use 150Gt/yr as the cutoff here? Is this the estimated balance discharge?

No, this was simply a rounded estimate (from 148 Gt yr-1) of the current discharge. This text has been removed.

p. 8, l. 15-21: The numbers presented in this section indicate that the flux is the areanormalized volume flow rate. Is this correct? I think that most readers would stumble in this section since it is not apparent from the start that the flux is normalized. I normally think of flux as a volume flow rate and I was perplexed by the seemingly contradictory statements in the first sentence until I looked at the flux units.

We used "volume flow rate" incorrectly when it was "mass flow rate". Mass flux by definition is area normalized with dimensions [mass time-1 length-2]. Flux is not flow rate. We explicitly use the term "mass flow rate" throughout the document when referring to a product with dimensions [mass time-1]. We have removed this section based comments from both reviewers.

p. 9, l. 19: The Enderlin et al. (2014) paper used bed elevations from radar picks. Examination of the original interpreted data for Koge Bugt suggest the bed was much deeper than the updated (and BedMachine) dataset. It is likely that the Bamber et al. (2013) bed map used the same radar data as Enderlin et al. (2014). (This comes up again on p. 11, l. 5.)

Thank you for clarifying and we have revised the text accordingly.

p. 9, l. 26: The use of Khan et al. (2016) surface elevation adjustments may also play a role since the Enderlin et al. (2014) elevations are directly extracted from DEMs and Operation IceBridge lidar timeseries.

Correct, and we have revised the text to reflect this.

- 3 Reply to Reviewer #2 doi:10.5194/essd-2019-29-RC2
- 3.1 Major Comments
- 3.1.1 Treatment of Velocity Data

On Page 10, line 5, the authors state that the signal to noise ratio may be reduced at the individual glacier level but that this noise is not apparent in the total ice sheet discharge time series. I disagree with this assertion and will highlight an example that I think demonstrates the need for a more careful treatment of outliers.

We agree with the reviewer that our use of unprocessed velocity data was not sufficient. In the revised text we now have a "Invalid Velocity" in addition to a "Missing Velocity" section and exclude outliers based on a standard deviation test.

We want to produce the best possible discharge product but at the same time we must rely on upstream data producers and cannot recreate them all from first principles. If we did, our results would certainly be less accurate than the existing products. We therefore have decided to do some additional processing of upstream products as you suggest (and as we already did for the ice thickness data). We also hope to make extensive use of the ESSD "Living Data" option and when updated velocity products are released, presumably each at higher fidelity, we will incorporate those. Unfortunately, it seems extremely difficult to develop a filter that flags noise but not real speed-up events. With velocity products now spanning shorter and shorter durations, they will begin to capture the noisy and dynamic true velocity of glaciers, which sometimes do exhibit 10 or 100 % speed-ups over short time periods. We also note that while you have highlighted a case where there is a low point between two high, in general the noise is high, and filtering reduces discharge. This implies that either the velocity errors are not evenly distributed, or true velocity increases are being filtered, because we know from glacier behavior that short-term changes are almost always speed-up events and not slow-down events.

The filter we selected (30 point moving mean, removing 2-sigma outliers, run 3x) appears to remove all outliers, but likely removes some real velocity spikes too. Our treatment reduces GIS annual average discharge by ~1% in most years, up to 4% in years with high discharge, and more in the 1980s when the data is noisy. We explain this in the revised text and add a section to the Appendix showing the same (filtered) time-series from the main paper text, but un-filtered.

Distinct spikes are apparent in the Greenland total graph (Figure 6) from the ~monthly data in 2011 and 2013. These seem to be primarily the result of similar spikes in the SE sector (Figure 7). In the SE in 2011, there is a rapid fall of ~50 Gt (30%) and a subsequent rise again of ~40 Gt all within a ~3-month period, which is physically improbable.

Looking more closely at the top 7 individual glacier plots in Figure 8, we see that the sector spike in 2011 is an artifact of discharge at Køge Bugt Glacier (located in the SE), which increases from a baseline of < 20 Gt/yr to over 40 Gt/yr in 1-2 months. Given the usage of annual surface elevations used in this study, this change must then be due to velocity, and would require an acceleration of over 100% in a very short time period. Below are velocity time series taken from Joughin et al. (2018) (https://doi.org/10.5194/tc-12-2211-2018), which shows no such dramatic acceleration during that time. From my quick glance at the individual glacier discharge data accompanying this manuscript, several other glaciers also occasionally exhibit large, abrupt changes that I suspect are untreated outliers.

Similarly, the next SE sector spike in 2013 is not present in either Køge Bugt or Kangerlussuaq Glacier, which are the largest two glaciers in this sector. This would mean that the remaining glaciers in the SE sector (which I estimate from the figure to roughly account for an average ~100 Gt/yr) would have to compensate for the ~50 Gt/yr increase. I'm skeptical of this because, to my knowledge, such an abrupt and short-lived acceleration in 2013 has not been previously documented.

These examples show that although signal noise at individual glaciers will typically be mitigated in the total ice sheet time series, large outliers at the more prominent glaciers will propagate to the sector-wide time series and also possibly to the total time series and impact annual averages. The outlier problem is amplified when linear temporal averaging is applied using noisy data points. I think the data quality could be improved by a simple low-pass filter, which would also make the individual glacier time series more robust for those who may use the data for local studies. It may be helpful to reference the velocity

maps/mosaics associated with outlying points to assess if the pixels are particular noisy at that time.

We agree and have applied a filter prior to the linear temporal averaging step.

**3.1.2 Temporal Averaging**

How are annual averages computed from the nonuniform time series? I suspect that the series are resampled at uniform intervals prior to averaging, but this is not explicitly described in the manuscript. Even if resampled at equal intervals, the use of linear interpolation for missing time periods means that there is an inherent sampling bias that the authors should estimate, though it may be small. This could be done using the reference period with dense temporal coverage.

We explored two temporal filling options:

- **1.** Annual verage of samples
- 2. Resample to daily resolution with linear interpolation, then compute annual average

We use method 2, similar to Joughin et al. (2018) and now state this explicitly. The difference between the methods is small in most years. The average across all years is < 5 % and the median < 3%. In theory and when using clean or modeled discharge estimates as in King et al. (2018), the error should be  $\sim 6\%$  at worst, because seasonal discharge variability is  $\sim 6\%$  (King et al. 2018).

We have updated the text to clarify this method and its possible effects.

3.1.3 Automated Flux Gate Selection Algorithm

I'd like to see more details on the algorithm included in the manuscript, especially since the automated algorithm is the key strength of this work. Some details are commented throughout the code samples, but it requires digging. I suggest a methods section describing the algorithm development that at minimum addresses:

**1 If and how frequent manual adjustments are needed due to continued retreat (terminus retreats behind 5km upstream of GIMP-determined terminus).**

No adjustments are needed because we set gates using the baseline period, years 2015, 2016, and 2017. There has been no retreat (that we have detected) > 5 km since then. We state that although the gates can be moved, and are for Fig. 2, they are not for the rest of the work.

**2 The treatment of unconfined, radially-draining catchments – do they require additional corrections as shape and direction of dominant flow change?**

No. The gate is defined 1x and, in this work, is stationary. It is true that over time some edge pixels might cross the threshold from greater than or equal to 100 m yr-1 to less than 100 m yr-1, and then slower ice would be included in the discharge estimate. Similarly, some ice initially excluded due to flow less than 100 m yr-1 may increase,

but not be included. We assume these edge cases are insignificant. Gate location and flow direction are independent. At each time step we calculate gate-orthogonal flow. This was described at the end of the discussion section (wrong location) in the submitted document. The text has now been moved to the methods section, but is again discussed briefly at the end of the discussion section.

**3 Brief description of treatment of floating ice shelves. By termini selection, do you mean grounding ice mask from BedMachine/GIMP or glacier front?**

**We have added text explaining that we are referring to grounding line mask.**

**4 the algorithm ever require "unfiltering" originally excluded pixels after gate migration? For example, though most glaciers accelerate toward the terminus, if a pixel 5 km upstream of a slow moving near-terminus pixel exceeds the 100 m/yr threshold, is it still excluded, or retroactively filled along the gate? I suspect this would only happen in some instances at radially-draining glaciers.**

Gates do not migrate. There is an additional case opposite to the one you describe above. If a fast terminus does not have fast ice upstream of it, the gate would not be positioned there. We have performed a limited manual inspection of the algorithm at each step and have observed one of these situations. I found one small glacier on the north west coast that had fast ice near the ice edge, but no fast ice inland, and therefore no gate. Given the size of this glacier - just a few pixels wide, I let the algorithm exclude it. Similarly, I examined the initial fast ice mask for locations where the mask does not reach the coast where I would expect it to. I did not see anywhere where this occurred. These two examinations were not exhaustive.

We are reasonably confident we have gates where anyone else ever placed gates manually, and then some more.

**3.2 Minor Comments**

Title and use of '2000-2018' period: The figures and the description that the period studied as '18 year' (abstract) indicate that the time series extends through the end of 2017 and excludes data from 2018. While the use of 2000 'to' 2018 could be taken to mean 'up until' 2018, '2000 to 2018' and '20XX to 2017' are interchangeably used throughout the manuscript and it is confusing to the reader which exact period is being referenced. For example, an average discharge from 2010 to 2017 is mentioned on line 6 of the abstract. Based on the usage of the title, is this taken to mean the average was calculated over the period 2010 through 2016, excluding 2017? Similarly, on page 7, line 27, a sector average is described over the 2007 to 2017 period. If the terminology is consistent and this does indeed refer to 2007 through 2016, why is 2017 excluded in these instances? Otherwise, please consider either replacing '2000 to 2018' with '2000 through 2017', or editing the remaining 'to 2017' references for consistency.

We apologize for the inconsistency and typos. We've decided that "through 2017" is the clearest phrasing and have changed all text to reflect this.

Page 2, Equation 1: Perhaps specify that A is area (even though it's intuitive) so that all terms are defined.

**Done.**

Page 2, Line 5: "... and Q is the volumetric flow rate". This makes is sound like Q should be in Equation 1. Are you defining now for later use? If the term is not used again it might be best to omit.

Q is in Eq. 1, but we've changed the text. Q is now the discharge flux (although the section of the text about discharge flux has been removed). Flux integrated over gate area equals flow rate.

Page 3, Line 2: Contribute should be "contribution"

**Done.**

Page 5, Line 20: The use of 917 kg/m3 density value should be noted again in the discussion when comparing to previous studies as it could be another, albeit small, source of difference.

We mention the density value we use twice in the methods, and once point out that others may use a reduced value if considering firn or crevasses. We don't see why anyone would use a value more dense than 917 kg m-3. This suggests the difference between our estimates and others who did use a reduced density is even larger than we show here.

Because there are so many different reasons others may have different estimates, we disagree with the need to explicitly mention density in the discussion, especially since that would not be likely to explain any differences, but rather increase differences.

3.4.1 Missing Velocity Do the reported stamps refer to the time span midpoint, or the first date of the time span (first image)?

We use the middle of the time spans when provided and now clarify this.

Page 6, Line 23: "This thickness adjustment adds 21 Gt to our baseline-period discharge estimate..." Should be Gt/yr?

**Yes, fixed.**

Is this adjustment, described in Table A2 and as applied to the final estimates in Figure 6, added as a fixed value to the full time series or does the magnitude of the adjustment vary through time ?

The thickness adjustment adds that amount relative to the unadjusted baseline discharge. We highlight this value to show the approximate impact of the thickness adjustment on discharge. The baseline discharge is never included in the time series (it is only used to place gates, there is no velocity product included elsewhere that spans 3 years). The 21 Gt yr-1 value is different at each time for the following two reasons (recall discharge  $\propto$  thick \* velocity): 1) the adjusted thickness is further adjusted

temporally with Khan et al. (2016), and more importantly, 2) velocity is variable in time.

3.4.3 Ice Discharge Uncertainty: Is the temporal variability in coverage considered in error estimates? I would expect discharge estimates for a given time with, for example, only ~20% coverage to have a larger uncertainty than a time point with full coverage.

Our uncertainty treatment is simplistic but conservative (because we treat all errors as systematic not random). You are correct that reduced coverage may have larger uncertainty, but it is also a function of time since any given pixel was last observed. 10% coverage with a 100% coverage one week on either side can be linearly filled with high confidence. 50% coverage with 100% coverage a few months away may have more uncertainty.

We do not quantify the effect of gap filling in the error estimate in units of Gt. We do provide the coverage in the data we release so that others can use coverage as they want - perhaps as a proxy for quality, uncertainty, remove all points < *C* coverage, etc.

4.2 Ice discharge (volumetric flow rate) Page 7, line 26: If 169 Gt/yr is 54% of total ice sheet discharge, this yields a total ice sheet discharge of ~313 Gt/yr. On line 28, 70 Gt/yr representing 32% of total ice sheet contribution would indicate that total ice sheet discharge is ~219 Gt/yr. These values are inconsistent with each other and with the preceding paragraph. Can some text be added here to clarify what these percentages represent?

**When calculating the contribution of a region to the total, we incorrectly removed that region from the total. This has been fixed.**

4.3 Volumetric flux: I found this section confusing. By normalizing discharge by crosssectional area, the authors are effectively describing interannual changes in velocity (since density is constant through time). I think it would be easier to follow if described in velocity terms, but it may not be necessary to include this section at all.

It should have been "mass flux" not "volumetric flux".

Density is constant but thickness is not, so this estimate is not describing only interannual changes in velocity. Because thickness changes slowly in this work, it is approximately describing intra-annual changes in velocity. Discharge mass flux (like discharge mass flow rate) is a function of both velocity and ice thickness. If the velocity remains the same but the ice thins by a factor of two, the discharge flow rate would decrease by a factor of two, but the discharge flux would remain steady.

After considering issues with this section raised by both Ellyn Enderlin and Reviewer #2 we have opted to remove it.

Page 8, line 18: 2013 maximum should be 2011 (as previously stated on page 7, line 24).

Flux is different than flow rate, and the year of maximum flow rate is not the same as

**the year of maximum flux. This section has been removed.**

Page 9, line 29: "The King et al. (2018) 2005 peak discharge is 524 + /-9 Gt dropping to 461 + /-9 Gt in 2008 – a decrease of ~63 Gt. In our work, the 2005 peak is 515 + /-50 Gt, dropping to 495 + /-50 Gt in 2008 – a decrease of only ~20 Gt."

This comparison should be altered to either (1) compare annual averages between both studies, or (2) compare absolute max and min value between 2005 and 2008 from the ~monthly estimates (which look to be about 550 - 480 = 70 Gt from Fig. 6). The annual changes from King et al. (2018), plotted in red in Figure 1 from that paper, show an annual change of ~20 Gt, which is comparable with these results.

You are correct, and we are happy to have it pointed out to us that our results are not significantly different from the other recent discharge estimate. We have revised this text.

Figure 2: The heatmap is an excellent addition to the manuscript and packs a lot of information into a very readable figure. Interesting to see sensitivity to cutoff velocities increases with upstream gate distance.

We are not sure why there is increased sensitivity to cutoff velocities with upstream gate distance. One hypothesis is that farther upstream, the slower velocities are more strongly influenced by SMB processes. That is, a 10 m yr-1 pixel 9000 m upstream would take 900 years to reach the ice edge (if it did not accelerate...).

Figure 8 and discussion of 7 largest discharging glaciers: I accessed the individual glacier discharge data available on the data portal and calculated that 'IKERTIVAQ\_M' glacier contributes a period average discharge of ~14 Gt/yr, which is larger than the average contribution from Nioghalvfjerdsbrae. Why was this glacier excluded from the top 7? If instead this is a reference to the top 7 glaciers spread throughout each sector rather than absolute largest 7, then consider adding a word on this for clarification.

The top five glaciers are clearly Sermeq Kujalleq (Jakobshavn Isbræ), Helheim, Kangerlussuaq, Køge Bugt C. and Zachariae Isstrøm. After that, it is sensitive to the definition of "top". We used the mean of the last year. Using the last point, the mean of the last year, the mean of the last three years, and the sum of the last five years all give different results. We now clarify the method in the text. We note that the gate labeled "Ikertivaq M" spans both the M and S sector from Mouginot et al. (2019) and appears as one large gate, while all other glaciers described here as the "top" are individual outlet glaciers.

Page 24, line 17: Is the gate number 263 or 264? Both values are used throughout the manuscript.

Updated and fixed due to updated input data.

Table A2: Are these values given for the reference period (2015-2017)?

Yes and clarified.

K. D. Mankoff

Kenneth D. Mankoff1, Willam Colgan1, Anne Solgaard1, Nanna B. Karlsson1, Andreas P. Ahlstrøm1, Dirk van As1, Jason E. Box1, Shfaqat Abbas Khan2, Kristian K. Kjeldsen1, Jeremie Mouginot3, and Robert S. Fausto1

[revised manuscript text omitted]
  $\frac{468}{438} \pm \frac{47}{43}$  Gt in 1986, has a minimum of  $421 \pm 42$  Gt in 1995, increases to  $452 \pm 45$  in 2000and 515-, further to  $504 \pm \frac{50}{49}$  Gt/yr in 2005, after which annual discharge remains

approximately steady at  $\frac{495 \text{ to } 520 \cdot 484 \text{ to } 503 \pm}{503 \pm} \sim 50 \text{ Gt/yr}$  during the 2005 to 2017 period. Annual maxima in ice discharged occurred in 2005 ( $\frac{515 \cdot 504 \pm 50 \cdot 49}{504 \pm} \text{ Gt/yr}$ ), 2011 ( $\frac{521 \cdot 499}{53 \cdot 50} \text{ Gt/yr}$ ), and  $\frac{2013 \cdot (519 \cdot 2014 \cdot (503 \pm 53 \cdot 51) \text{ Gt/yr})}{53 \cdot 51} \text{ Gt/yr}$ ). Discharge in both 2016 and 2017 was less than 500 Gt each year.

At the sector region scale, the SE glaciers (see Fig. 1 for sectors regions) are responsible for  $\frac{148 \text{ to } 169 - 139 \text{ to } 167 (\pm 12 - 11)}{120 \text{ to } 167 (\pm 12 - 11)}$ %) Gt yr-1 of discharge ( $\frac{42 \text{ to } 54 - 30 \text{ to } 34}{30 \text{ 
[revised manuscript text omitted]

**Figures**